# Impact of overdose prevention sites during a public health emergency in Victoria, Canada

**Bernadette Pauly**[1,2‡]*, **Bruce Wallace**[1,3‡], **Flora Pagan** [1‡], **Jack Phillips**[4°], **Mark Wilson**[4°], **Heather Hobbs**[5°], **Joann Connolly**[6°]

**1** Canadian Institute for Substance Use Research, Victoria, Canada, **2** School of Nursing, University of Victoria, Victoria, Canada, **3** School of Social Work, University of Victoria, Victoria, Canada, **4** SOLID Outreach, Victoria, Canada, **5** AIDS Vancouver Island (AVI), Victoria, Canada, **6** Victoria Cool Aid Society, Victoria, Canada

° These authors contributed equally to this work.
‡ BP, BW are joint senior authors. FP assisted with analysis and conceptualized the findings.
* bpauly@uvic.ca

**Data Availability Statement:** Data may be available upon request as there are ethical restrictions on sharing it publicly. The data contains confidential information and are restricted by the University of

## Abstract

The primary objective of this study was to examine the impacts associated with implementation of overdose preventions sites (OPSs) in Victoria, Canada during a declared provincial public health overdose emergency. A rapid case study design was employed with three OPSs constituting the cases. Data were collected through semi-structured interviews with 15 staff, including experiential staff, and 12 service users. Theoretically, we were informed by the Consolidated Framework for Implementation Research. This framework, combined with a case study design, is well suited for generating insight into the impacts of an intervention. Zero deaths were identified as a key impact and indicator of success. The implementation of OPSs increased opportunities for early intervention in the event of an overdose, reducing trauma for staff and service users, and facilitated organizational transitions from provision of safer supplies to safer spaces. Providing a safer space meant drug use no longer needed to be concealed, with the effect of mitigating drug related stigma and facilitating a shift from shame and blame to increasing trust and development of relationships with increased opportunities to provide connections to other services. These impacts were achieved with few new resources highlighting the commitment of agencies and harm reduction workers, particularly those with lived experience, in achieving beneficial impacts. Although mitigating harms of overdose, OPSs do not address the root problem of an unsafe drug supply. OPSs are important life-saving interventions, but more is needed to address the current contamination of the illicit drug supply including provision of a safer supply.

## Introduction

Overdose deaths have been escalating in North America for over a decade [1–3]. The illicit drug overdose crisis in North America has had devastating impacts on individuals, families and communities, including premature loss of life and even lowering life expectancy [4, 5]. In Canada, there has been an increase from 8.4 overdose deaths per 100,000 residents in 2016 to

Victoria. Data access requests may be made to ethicschair@uvic.ca.

**Funding:** This research was supported by Bernadette Pauly's Island Health Scholar in Residence Award, Island Health. The funders had no role in the study design, data collection and analysis, decision to publish, or preparation of the manuscript.

**Competing interests:** The authors have declared that no competing interests exist.

12 per 100,000 in 2018. The province of British Columbia (BC) experienced the highest rate of overdose deaths in the country with 1,525 deaths in 2018, and a dramatic rise in overdoses from 5.9 overdoses per 100,000 residents in 2012 to 30.3 per 100,000 in 2017 [6]. This high rate of overdose deaths continues unabated, with an estimated four deaths per day in BC. In April 2016, due to dramatic increases in overdose deaths, the BC Provincial Health Officer declared illicit drug overdoses a public health emergency [7, 8]. Three years later, this state of emergency remains in effect. Overdoses are the top cause of unnatural death in the province, with illicit fentanyl detected in 87% of drug overdose deaths [6].

To prevent overdose deaths, there are ongoing calls internationally for the rapid scale-up of supervised consumption services [9–11]. Supervised consumption services (SCSs) are a harm reduction intervention that helps to mitigate the harms of drug use, through onsite monitoring and rapid intervention by trained staff in the case of an overdose. There is a robust evidence base for SCSs [11–13]. These services have been found to reduce overdose deaths, reduce transmission of blood borne diseases such as HIV and Hepatitis C, and connect people to other services (e.g. primary care, detoxification and withdrawal services). In spite of the evidence, SCS implementation has been heavily regulated, cumbersome and suppressed in Canada, including throughout BC, due to federal requirements such as feasibility studies, police approvals, community consultations and delayed approval processes [11, 14–16]. At the time the public health emergency was declared, there were only two federally-sanctioned SCSs in Canada, both located in Vancouver, BC.

In December 2016, as a response to the ongoing and rising rate of overdose deaths, the BC Health Minister issued a ministerial order that directed all regional health authorities to provide overdose prevention services as ancillary health services and as a provincially-sanctioned, extraordinary measure [17]. Overdose Prevention Sites (OPSs) originated from grassroots activism and drug user organizing with established "pop-up" unsanctioned sites in a few locations in BC prior to the 2016 Ministerial order. This history of activism is similar to that preceding the establishment of other harm reduction services by people who use drugs in BC, including needle exchange, and supervised injection [18–20]. The rational for grassroots OPSs was in part driven by the widespread promotion and acceptance of take-home naloxone (THN), public health advice not to use illicit drugs alone, and the limited availability of SCSs in the province underscoring the lack of safe spaces for use.

Overdose Prevention Sites (OPSs), like SCSs, provide spaces for people to inject previously-obtained illegal substances with sterile equipment, in settings where they can be observed and others can quickly intervene in the event of an overdose. OPSs were established as part of the BC public health emergency response, to provide monitoring and rapid intervention in the case of an overdose. Generally, OPSs are staffed by experiential (people with lived experience of drug use) and non-experiential harm reduction workers and are provincially rather than federally-sanctioned as a temporary emergency measure. This distinguishes them from supervised consumption sites (SCSs), which are federally approved and with most often having nursing staff on site. In the year following the ministerial order sanctioning OPSs, twenty-five OPSs were established in BC, with 545,488 visits [21]. At least 2,500 overdoses were responded to, and many more prevented, with no fatalities at any site. At the same time, the federal government of Canada introduced new legislation to speed up approvals of permanent, federally-sanctioned consumption sites with approvals of seven new sites in the province [22]. Several OPSs have transitioned to federally-sanctioned sites. Currently, there are 33 supervised consumption spaces in BC [23]. Both OPSs and SCSs have contributed to overall reduction in morbidity and mortality in the province [6, 24].

Compared to SCSs, there is limited research on OPSs as community-led grassroots initiatives. OPSs have been identified as a provincially-sanctioned but community-led response that

is novel and nimble, providing an alternative to cumbersome federally-sanctioned processes [25]. In a Vancouver study, three mixed gender OPSs were described by women as providing safer spaces for women in the context of increased vulnerability to overdoses and gendered violence. Additionally, peer to peer OPSs were perceived to increase comfort and safety for those accessing services, though experiential workers (peers) were often underpaid and their work unrecognized [26].

The overall focus of our research was to explore the early implementation and impacts of OPSs through a rapid case study design. Findings related to early implementation are reported elsewhere [25]. In this paper, we explore the perceived impacts of implementing OPSs in the context of an illicit drug overdose emergency, by those who access and provide services in one city (Victoria, BC) with a high rate of overdose deaths during a declared public health emergency. This research may provide useful insights for other jurisdictions.

## Materials and methods

To facilitate rapid conduct of the research, we did not undertake a fully participatory research project. However, we did draw on principles of collaboration that underpin community-based participatory research [27–29]. At the time of the public health emergency and the establishment of the OPSs, the academic researchers (BP, BW, FP) were already highly engaged with community organizations and groups responding to the overdose emergency and had long standing research partnerships spanning many years with several of the agencies [30, 31]. The need for OPS research was identified as a community priority during the implementation process, and community members were actively involved in the research. Victoria is the capital of BC and one of the top three townships in the province impacted by overdose deaths. As well, the city has a history of documented challenges related to the establishment and expansion of harm reduction services [32–34] even though the province of BC is known as a world leader in harm reduction with the most robust and progressive provincial harm reduction policies in Canada [35, 36]

There were four potential OPSs in Victoria, All four were contacted, with three agreeing to participate in this research project. A fourth agency expressed support but was unable to facilitate the research in the agency in the short timeframe. In addition, the local drug user organization was approached, and agreed to participate as an agency contracted to staff OPS sites. Staff from the three organizations and the local drug user agency formed a research advisory. This group met at the beginning and throughout the study to assist with design and conduct of the study, reviewed and assisted with writing of research findings.

Research Design: Within this collaborative approach, we employed a case study design. Case study research allows for in depth and detailed study of phenomena within a real-life context [37]. In this project each OPS constituted a case, and our focus of study was the process and impacts of implementing OPSs during a public health overdose emergency. Table 1 provides a description of the three OPSs. With three cases, we met the criteria described by Stake for strong cross case comparisons [37]. Using cross case comparisons, we examined similarities and differences in implementation which are reported elsewhere [25] and impacts reported here.

Theoretical Framework: Our study was guided by the Consolidated Framework for Implementation Framework (CFIR) [38]. As one approach to implementation research, the CFIR aids investigation of the complexity of implementation processes by exploring the influences of five factors, or domains: intervention characteristics, outer setting, inner setting, characteristics of the individuals involved, and the process of implementation. This framework was utilized to inform and construct the interview guides as well as guide the analytical process. Both

Table 1. Overdose prevention sites case descriptions.

| | OPS #1 | OPS #2 | OPS #3 |
|---|---|---|---|
| Location | Embedded in Homeless Drop-In Programme | Embedded in Harm Reduction Drop-In Programme | Emergency Homeless Shelter |
| Staff[1] | Trained experiential staff Trained Emergency Staff | Trained harm reduction workers (including experiential workers) | Trained experiential staff Trained harm reduction workers |
| Capacity | Three stainless steel tables No chill out space | Four stainless steel tables Chill out area in drop in space | Three stainless steel tables Chill out area outside injecting room |
| Method of Consumption | Injection only | Injection only | Injection room and inhalation tent |
| Operating Hours | 7 days a week/14 hours | 7 days a week/6 hours | 7 days a week/14 hours |
| Equipment | Naloxone, Harm reduction supplies (clean supplies for smoking and injecting) | Naloxone, Harm reduction supplies (clean supplies for smoking and injecting), Oxygen and pulse oximeter | Naloxone, Harm reduction supplies (clean supplies for smoking and injecting) |
| Other Services Available | Meals, Access to showers, Clothing, Daytime programming, Counselling, Outreach, Support and referrals | Meals, Access to harm reduction supplies and education, One-on-one and group support, Public education, Advocacy | Meals, Access to harm reduction supplies, 30 day stay, Daytime programming, Case management, Showers, Clothing, Ministry worker, Counselling, Primary care including OAT prescriptions. |

[1] Experiential Workers: We would note that the use of the term 'peer' to refer to people who use drugs is contested and a term more likely ascribed by institutions to people who use drugs than an identifier chosen by people who use drugs themselves. In the absence of political self-identifiers, we will refer to people with lived/living experience (PWLE) and in the context of work, experiential staff or workers. Training includes Naloxone training primarily.

Trained Harm Reduction Workers: most often non-experiential workers with formal education. At OPS #2 the title Harm reduction workers did include experiential workers.

Trained Emergency Staff: Health Care professionals with formal emergency.healthcare training

case study designs. as described by Stake [37]. and the CFIR framework seek to understand the context and influences on specific phenomena: in this case the early implementation and impacts of OPSs. Data Collection: Purposive sampling of a minimum of three staff from each agency was facilitated by the primary contact person at each agency. We sought to include an "implementer", a person identified by the agency as involved in the implementation of the service, staff persons working within the service who identified as 'experiential' and 'non experiential'. Recruitment emails or handouts were provided by the agency's contact person to prospective participants. Informed written consent were signed following a verbal and written overview of the research, and $20CDN stipends were provided for experiential workers and those who completed the interview outside of paid hours.

Participants accessing the OPS services were given handbills by staff, most often the experiential staff, to invite people using the services to participate. At each site, a minimum of three people using the service were recruited. At the busiest site, the initial three people recruited all identified as male. Thus, three additional service users identifying as female were purposively recruited. Informed written consent was obtained following provision of a verbal and written overview of the research. A stipend of $20CDN was provided at the start of each interview. All interviewes were conducted at each agency in a private room. Interviews were recorded and transcribed verbatim by an experienced transcriber associated with our research centre.

Qualitative semi-structured in-depth interviews were conducted by the academic researchers (BP & BW) across the three sites with 27 participants. Most interviews lasted just over 30 minutes. . .Data were collected within the initial two to three months of operation of the services during a five week period from March to April 2017. Ethical approval for this study was obtained from the University of Victoria Human Research Ethics Office (17–032).

Sample: Fifteen interviews were with harm reduction staff, including workers with lived/living experience and 12 interviews were with people who use the services. Demographic characteristics of participants are given in Table 2.

## Analysis

All of the interview transcripts were entered into NVIVO for coding. As a first step in the data analysis process, initial transcripts were reviewed by FP, BP, BW and coded inductively looking for broad, salient concepts or ideas. These were then discussed as a group to develop initial coding categories. FP used the coding framework to continue coding with FP, BP, and BW meeting to further refining the coding categories in an iterative manner moving between data and codes. With the assistance of NVivo software, FP continued the process of coding and analysis. With all data coded and themes refined for each case, we utilized cross-case comparison [39] to examine similarities and differences between cases. We compared and contrasted themes across the three cases to develop cross-case conclusions about the impacts of OPS. The CFIR Framework was used to further interpret the findings. Findings were reviewed with agency partners who provided clarification and confirmation of findings after case

**Table 2. Participant characteristics.**

|  | N (%) |
|---|---|
| **Staff** | **15** |
| Average age | 38.7 years |
| Gender* | 9 (60%) Female |
|  | 6 (40%) Male |
| Ethnic group | 12 (80%) White |
|  | 2 (13%) Indigenous |
|  | 1 (7%) Other |
| Education | 10 (67%) College or university |
|  | 5 (33%) Some high school or completed |
| **Service users** | **12** |
| Average age | 42.3 years |
| Gender* | 6 (50%) Female |
|  | 6 (50%) Male |
| Ethnic group | 7 (58%) White |
|  | 4 (33%) Indigenous |
|  | 1 (8%) Refused |
| Education | 4 (33%) Some college or university |
|  | 2 (17%) Technical school |
|  | 3 (25%) Some high school |
|  | 3 (25%) Elementary |
| Housing | 6 (50%) No fixed address |
|  | 4 (33%) Shelter |
|  | 2 (17%) Supportive housing |
| Stable housing | 1 (8%) |
| Long term city resident | 9 (75%) 10 years or whole life in city |
| Primary source of income | 5 (42%) Disability assistance |
|  | 5 (42%) Social assistance |
|  | 2 (17%) Employment income |

*Participants were asked "What gender do you identify with (e.g. male, female, transgender or 'other')?

descriptions were developed and once cross case analysis had been completed. Findings were also shared with the regional health authority and a federal agency.

## Findings

### From trauma to zero deaths

Prior to OPS implementation, washrooms at all three agencies were used for covert substance use by clients, with overdoses happening more and more frequently behind closed doors as fentanyl related overdose deaths continued to rise. Each agency dealt with on-site substance use in different ways at different points. This included intentionally overlooking "discrete" substance use on-site; limiting access to services for those found using or having overdosed on agency premises; and for one agency, developing protocols to monitor washrooms and respond to overdoses.

Despite differences, participants at all sites had similar descriptions of how the opening of the OPSs impacted staff in all three agencies. Participants expressed that it was a relief to not find people slumped over, blue, in bathrooms, and then fighting to revive them not knowing how long they had been there. Similar to other research we conducted, they described the situation pre-implementation as one which was traumatic for both staff and service users [40]. This participant discusses the relief experienced by staff and service users:

> I think it's created a huge relief for a lot of [staff]. And for [service users] as well because, I mean, we were getting to a point where we were responding to overdoses on a daily basis. And so, shifting from outreach workers to emergency response on a daily basis was putting a toll on all of us. Because not only could we not do our jobs, [service users] weren't feeling supported as much anymore because we're just putting out fires. Consistently. So, I think that kind of takes a toll on everybody. [Staff 05]

From the perspective of all participants, the most agreed-upon positive impact across all three sites was the fact that zero deaths had occurred in any OPS. When asked directly about successes, one staff person stated "*we're saving lives, that's successful*" [Staff 15]. A service user at the same site similarly noted "*people are being saved, that's the bottom line*" [Service User 03]. This service user goes on to note how critical OPSs are to saving lives.

> It's, been so helpful to me, like very helpful. Because I know I can use and I don't have to worry about OD'ing and being by myself . . .I have that reassurance, because that's always going through my mind, like "Oh my god, is this going to be the last time I shoot? Like, I've got to say goodbye to people." And like, it was just hard on me because I was thinking about my friends, and they certainly never thought that they weren't going to pull through. [service user 14]

'Zero deaths' is consistent across both British Columbia and Canada, as well as internationally, with no deaths occurring in any sanctioned injection witnessing or supervised facility. In this study, participants defined OPS success according to the metric of zero overdose deaths. The opening of the sites had the impact of being able to openly save lives and relief at being able to prevent overdoses instead of the trauma of finding people unconscious in bathrooms.

### From overdose crisis to overdose prevention

With the implementation of OPSs, experiential and non-experiential staff in all agencies were able to monitor service users and thus, intervene earlier to prevent overdoses than previously.

For example, staff described having the opportunity to provide physical stimulation to service users "nodding", or rubbing their arms or chest, to avert overdose:

> *I think a big thing that I didn't anticipate about the overdose prevention sites is catching the overdose before it's happening, you can totally do that. Like, I thought it would just be like people dropping, dropping, dropping, but if you know them and you keep an eye on them you can just tell like if I leave it one more minute they're going down. So, you can like get them up, get them breathing, sternum rub, give them water, and you actually prevent the overdose.* [Staff 18]

At one site, staff had use of a pulse oximeter to monitor oxygen levels, and could provide oxygen before levels dropped significantly. This meant an overdose could be averted without the use of naloxone as this staff person expressed: "*we've responded to overdoses . . .without naloxone, that I think we probably would have had to use naloxone if we were in our washroom setting*". [Staff 06]. Another staff at the same site added:

> *. . .that's a lot safer than going into a bathroom where somebody has been unconscious and you have to engage in a more invasive response which includes injecting naloxone into somebody's body, giving rescue breathes, and should their heart rate go, should they lose their cardiac functioning do chest compressions. And so, it's a much more preventative, responsive way to deal with the overdose.* [staff 01]

With the implementation of OPSs, staff at all agencies reported being able to intervene earlier, with reduced need for naloxone as a response to reverse overdose. Preventing overdose emergencies was attributed to being able to witness drug use, where previously drug use was hidden and often concealed behind closed doors. With the OPSs, service users were monitored and could receive support to prevent overdoses. This reduced the need for emergency responses, including reduced ambulance calls. Thus, introducing OPSs shifts who is the first responder and mitigates the risk of overdoses driven by the outer context (e.g. changes in the illicit drug market) by modifying the inner setting to be a safer space for use.

### From safer supplies to safer spaces

Two of the 3 agencies provided safer use supplies prior to OPS implementation. One staff person described the limitation of providing safe supplies without a safer space to use: "*Instead of saying 'We're welcoming you, this is a warm, safe environment to go,' we were saying "Go into the street where you're going to get further shamed by the public, and then come back"* [Staff 05]. A service user described the impact of accessing the OPS in an agency that did not previously have harm reduction services: "*well, I don't need to go in the bathroom and be told 'leave for a week' and stuff like that. You know, it's [the OPS] a plus for me for that."* [Service User 02]. At a site that tolerated "discrete" use a staff person explained that "*we gave out gear, but it was always with a 'don't use publicly' kind of thing*" [Staff 27]. Even staff at the agency that tolerated 'discrete' use commented on the significance of providing space:

> *We practice[d] harm reduction before, but at the same time we were telling people going to the washroom, "I don't want to see your drugs, I don't want to see anything", it wasn't a very healthy harm reduction approach, so now we have the opportunity to have the space where we invite people in and say that "your drugs are fine in here, you can come and use them however you need and however you see".* [Staff 06]

The implementation of OPSs highlighted the importance of providing a sanctioned space on-site for substance use as a harm reduction practice. The OPSs provided space to be observed, but also time to test their drugs and 'chill out' afterwards. One staff person described:

*"We've noticed people, before, they would go to the bathroom and they'd try and do their hit as quickly as possible to not get found. And here, they'll come into the OPS, and they'll use a smaller amount of it to make sure that it is what it is and that they can handle it, and then they'll use the rest. And they know that they have that time, they're not going to be rushed out of the spot. Because when people are rushed, that's when they tend to overdose.* [Staff 12]

Service users and staff also commented on how OPSs operated to address and respond to concerns related to public injecting as one service user explained: *"people don't have to see other people using out on the streets and stuff."* [Service User 25]. This speaks to value of a safer space for users and the public.

The shift from safer supplies to safer spaces exposed limits and contradiction of distribution of supplies only. Elsewhere, we have described this as incomplete harm reduction [40, 41] because individuals have access to supplies but are forced to use in unsafe spaces. Above the participants highlight mechanisms by which OPSs, like SCSs, prevent overdoses through not using alone as well as time and space to test drugs and less rushed injecting.

### From concealing use to destigmatizing use

By providing space for the preparation and use of illicit substances, service users no longer had to conceal their drug use or fear being caught using substances. The OPS not only provided an alternative to unsafe and public injecting but reduced the risks of arrest:

*It's a lot better to inject in the OPS . . ... Like cops come, they can either charge you or arrest you for, you know, using out in public and stuff like that. We try to hide it as much as we can, but it's not always hide-able.* [Service User 24].

Further to this, participants described how no longer having to conceal activities also meant people could be more honest about their substance use. An experiential worker stated:

*. . . people are able to be more honest and they don't feel like this place is a . . . place to fuckin' go hide in shame and do their drugs. Maybe it is a bit more empowering, 'cause you know they're talking to people who, you know, seem to care about them?"* [Experiential Staff 21].

Another experiential worker related *"you mean, I can leave my house and go do my drugs somewhere and be treated normal*?", stating: *"That's huge. That's a whole new thinking on that, you know*?" [Experiential Staff 08]. A non-experiential staff person, who had a long history of working to establish an SCS, was surprised by the impacts associated with opening the OPS on stigma:

*We knew how important it was to have supervised consumption. . . basically saying "come in with your drugs", "you're welcome here with your drugs, you can have them out, show me them" . . . you know, "your drugs aren't bad anymore". As much as we wanted to say that, we didn't practice those things, and again we have an amazing staff and we don't promote stigma and we do not shame people for being drug users . . . but there was no way around it in some of those scenarios.* [Staff 06]

This worker brings to the fore that not being able to provide a safe space inadvertently and unintentionally perpetuated stigma. Others highlighted how providing a safer space and reducing drug related stigma opened up new possibilities for assistance. A staff person related that "*I had someone in the first week that we opened ask us for help and he was like, "that's the first time I've ever felt comfortable asking for help in my entire life", and he was like a 45-year old male. So, I think people were just happy to feel welcomed in this space, while injecting, cause that's such a new idea, unfortunately.*" [Staff 16]

However, the transition from hiding use to discussing use openly was at times difficult in some sites due to the nature of prior stances on drug use and relationships between participants. As this staff person described: "*at the beginning, people would come in and be like 'I can't do this [laughing a bit] with you guys in here, and watching me.'. . .So we've kind of slowly been getting over that*" [Staff 12]. At the same site, an experiential staff person described the impacts of the shift from staff penalizing drug use to now accepting and witnessing it:

*At first the clients were like terrified. You know, these people kicked them out of bathrooms and barred them for weeks and months on end because of drug use and that kind of thing. Now, suddenly they're allowed to sit in this room, use drugs openly in front of the people that actually chastised them for years, you know? So it was pretty tricky, it's been really tricky trying to get them used to the idea that it's okay and it's safe to be there, and it's allowed to be there because it's actually the safest way to go. And that they actually, [the staff] wants them to be there and doing it this way because it's safer. And so it's taken quite a while for them to get used to this idea.* [Experiential Staff 08]

Many described the OPSs as significant in breaking down stigma through shifts in the inner setting. However, as an intervention, participants reminded us that stigma amongst people who use drugs and service providers is one component of a larger set of issues in the inner and outer contexts. For example, establishment of OPSs did not necessarily address the issues of banning people from a service, particularly for behavioral issues. One participant stated: "*But the problem is, people who are barred from [agency name] are barred from the [OPS] as well*" [Staff 12]. Provision of safer space requires that organizational policies be addressed so as not to restrict access. As well, there is the larger issue of criminalization in society. Another staff participant described:

*Overdose prevention sites are ways that we can break down stigma and another way that we can reinforce to people that their lives matter and that they have the right to be able to easily access basic health care, and be treated like human beings. And, that's not insignificant, but it's a really small part of the larger puzzle and it really does feel like a crisis response to a crisis situation and not a comprehensive response to a really big problem which, is ultimately drug policy, both federally and globally.* [Staff 01]

This participant reminds us that OPSs are an important intervention for countering stigma and preventing overdose deaths, but that OPSs alone are not a complete response to harms of substance use given factors in the outer context. OPSs are part of an emergency response and, as this participant highlights, ultimately the problem of overdose implicates current drug policy and the lack of an unsafe and unregulated source of substances.

### From shame to trust and relationships

Linked to the reduction of drug related stigma related, service users reported increased feelings of trust with agency staff. An experiential worker described the shift during OPS

implementation as "*it's starting just now, to start to branch out from the general, 'I come here to use, to use safely' and that kind of jazz, to [service users discussing] what's going on in their lives and that kind of stuff*" [Experiential Staff 08]. The provision of a safer space simultaneously provided more time for both the person to use their drugs, and increased contact with staff:

> *I just feel like I can sit and actually, like, talk to them for 20 minutes, 15 minutes, while I'm doing my shot. And most of the time its five minutes, like 'Hi, how are you?' you know? They've just been really super supportive because they know now what I'm going through. They didn't know what I was going through in the bathroom and they were like 'Come on, there's a wait,' you know, 'Your 15 minutes is up.' . . . So now they are like '[participant name], do you want the heating pad? Like to get your arms, get the blood circulating? Do you want some help finding a vein?' And that's so nice, like I wouldn't be able to do it before. I never had access to a heating pad. And I just feel like it's so much cleaner, it's so calm. I don't have somebody banging on the door, like 'Hurry up! We need to use it.' I don't feel rushed. Like it's just a lot calmer feeling. So I like it a lot better.* [Service User 14]

At one agency, service users described wanting to visit the OPS even when they did not need to inject, as they were able to spend time with staff including experiential workers. Another participant highlighted the symbolic importance of providing safer spaces as a way of building trust:

> *People who use drugs have such serious trust issues and rightfully so, throughout their lives I think they have been marginalized by people who help, and to actually invite them in and say 'your drugs are okay here, please, please tell me about them, show, show, show me them, show me how you inject', you know? 'Teach me'.* [Staff 06]

Participants also described increased intimacy that contributed to the development of trust:

> *There's a certain intimacy level between people who are using the services and staff. There's a certain intimacy around witnessing somebody using and that being okay, and that being the specific role for that space, to provide a safe, non-judgmental place for people to use and so, even folks that we've had really solid relationships with for quite a while, who already have quite a high level of trust in us, have opened up that much more, with staff, and engaged in much more of an intimate conversation about how they are doing, what their goals are, because they're calm and they feel really reassured and they feel non-judged, non-judgment, because they are doing something that is so highly stigmatized in a safe place where we're saying it's okay to do this.* [Staff 01]

Trust was enhanced between staff and service users as they were able to spend more time together and after sharing what can be an intimate and personal activity. This finding of increased trust and relationships applied to all staff with the implementation of OPSs. At two of the OPSs, participants specifically highlighted the important role of experiential workers in enhancing trust. One experiential worker described the importance of having people with lived experience being involved early in the design phase:

> *The [agency] came to ask us if we could help to build an atmosphere in which, that their service would be utilized best. . ..So we actually went on the ground for two weeks before it opened to have direct conversation with the population to make sure that they would be utilizing the service when it opened.* [Experiential Staff 13]

As this participant points out their involvement ensured that the service would be designed to be user friendly. When services were implemented, participants made clear that experiential workers had a critical role in OPS operations:

*Having the peers there makes it a success. It wouldn't be if it was just [OPS name] employees . . . users get to see a friendly face instead of an authority figure. . .someone who can save their life and who is on the same level. . .someone that talks to them differently than staff do, more knowledge about drugs, more empathy because people are friends, going to try harder to make sure people aren't going to die because they're not just a "nobody". . .even though staff are really good.* [Experiential Staff 17]

People with lived experience played a central role in enhancing access to OPSs in both the design and operation of these services. Experiential workers were specifically identified as helpful to establish trust in the service if they were known to services users. This highlights the importance of lived experience, in achieving important and often unattended to outcomes such as trust that mediate use of services [26].

## Financial, spatial and temporal challenges

While participants readily stressed the beneficial impacts of OPSs, they also highlighted challenges, particularly in the outer context, associated with implementing a new service with few new resources. Participants identified that the successes were achieved with existing or limited new financial and human resources, in existing spaces transformed for use as an overdose prevention site. Here, we examine the outer context and the challenges associated with implementing OPSs in a context of scarcity. This further highlights the commitment, creativity and adaptability of harm reduction workers (including those with lived experience) to make the services successful.

OPSs were implemented under a Ministerial order which provided a directive to establish services. As described elsewhere [25] and here, this meant that services could be developed by community agencies. However, given that they were implemented rapidly as an emergency measure, this also meant uncertainty related to financial resources. In some cases, OPSs were quickly established with few new resources. Where funding was available, it was often provided for limited periods making the future of an OPS uncertain as this staff notes: "*We only have funding right now[un]til [four months from now], so that's an interesting challenge cause neither me nor the residents, nor the workers . . . know if we'll be open past [then]. . . which is a bit challenging*" [Staff 16]. While there were specific funds dedicated to the overdose emergency from both federal and provincial governments, accessibility, availability and permanency of these funds were always in question.

Given that OPSs were implemented rapidly and in existing spaces and often with limited funding, there were challenges related to the physical spaces and the capacity of sites to handle the obvious need and demand for these services. Participants observed that a barrier to using the service was limited hours of operation. A service user stated: "*the last bathroom is at 9 o'clock though. So we're kind of forced just to stay outside*" [Service User 24]. The hours of service do not always align with service user needs, forcing people to inject in unsafe spaces. Furthermore, spaces were only able to accommodate three to four people at a time. In one site, set up was in a temporary trailer, there was no 'chill out space', and the space was often described as 'overcrowded'. One service user related:

"*. . .so when I do get really high I would rather just sit in the corner and do my own thing. Draw, or write, but when there's so many people are sitting around, it's hard to do that. You*

*know when someone is bumping the table when you are trying to color, and it's like 'alright I'll let that one go' and then five more times it's pure aggravation and then I lose my high, which is, then I have to have to do it all over again. . . I can't afford that I'm homeless. . .* [Service User 10]

Further, there were spatial issues related to ventilation which were identified as concerns related both to cooking as well as smoking substances.

Staff and service users all identified that in addition to the physical spatial challenges, there were temporal challenges, specifically wait times. One service user described: *". . .[staff are] just mostly all sad that there's not enough space for people, always sending people away. They're like, "Oh, just a couple of minutes, just a couple minutes." I'm like, "Minutes kill," you know, literally."* [Service User 04]. The same participant described using in the bathroom as the OPS was full, relaying *"It's going to be at least ten minutes. And when you're sick ten minutes might as well be five hours, you know, there's no waiting.* As several participants pointed out, it can take 10 to 20 minutes to find a vein.

*But that's another hurdle to having a small space and having a lot of users, is it takes some people 20 minutes to find their vein, so can you like fairly say "I don't care if you didn't find your vein get out"? . . .That's definitely a struggle that I found with just being in a smaller site, cause you know sites that have 7 or 8 chairs it might not happen as much..* [Staff 18]

The lack of space in the midst of unmet demand meant that people may return to injecting in unsafe situations: *"There's still people who come every day and we have to tell that there's a lineup to, and that they still go back to the bathroom, or out to the street, or wherever they go."* [Experiential Staff 13]

Another issue raised related to space limitations was the lack of privacy. This included lack of privacy related to the intimate act of injecting while under the watchful gaze of someone else, as well as the lack of privacy for confidential conversations. The lack of physical privacy was a specific concern for women, in a space accessed by men, as described by the female participant below:

*I didn't feel comfortable at first. Um, just because I have to inject in my legs and stuff, I've run out of veins everywhere else. So it's kind of, I don't want to pull my pants down in front of everybody, right?* [Service User 24]

Although not directly related to resourcing, participants did highlight that OPSs are reaching some populations better than others. Women, youth, those who identify as trans/non-binary, and all people outside of urban street-based settings were all mentioned as populations who had less access to OPSs:

*I think there's definitely people who don't necessarily feel . . .comfortable coming to a service like ours because it's fairly street level. . .. like there is definitely like . . .drug users who are less, visible and maybe are housed and are using in their own homes and may not feel the need or want to come to a service like ours.* [staff 01]

This was highlighted as a gap in service and that there is a need for additional services that would reach a broader population of people who use illicit substances. As well, participants highlighted that existing OPS services were important to connect people to other health and social services such as primary care, detoxification and withdrawal, and social services.

However, these connections were often limited as OPSs, unlike SCSs, generally do not have health care professionals, particularly nurses or social workers, available as a resource or onsite.

## Limitations

There are a number of limitations to this study. The study was conducted in a single city and the sample size does not reflect the complete range of OPS service designs, or experiences and perceptions. Our sample did not fully capture views of youth, women, LGBTQ and Indigenous participants. The research was conducted during the early implementation phase and reflects the origins of the services, and not how they have evolved over time or the effect on staff (experiential and non-experiential) of working in an OPS and cumulative effect of reversing overdoses overtime. Our research presents the perceptions of those accessing and providing services within these OPSs, and does not capture the perceptions of people who do not access the services who may have very different perceptions than what we heard. Despite these limitations, the study captures a unique moment during the rapid implementation of an international and pioneering harm reduction response, within a context of a public health overdose emergency.

## Discussion

The rapid implementation of OPSs in British Columbia, Canada, demonstrates an unprecedented expansion of government sanctioned consumption services. It provides an example with international relevance of an alternative to the restrictive sanctioning processes for supervised consumption services (SCS) [25]. According to participants who accessed and staffed OPSs in one city, the impacts of implementing this novel response to overdose were significant.

Similar to SCS globally, OPSs in BC have facilitated hundreds of thousands of visits annually with zero overdose deaths. In this study, zero overdose deaths was a key impact and indicator of success. Other impacts include shifting from a crisis response to preventing overdoses and reduced trauma, shifting from safer supplies to safer spaces as an alternative to banning use, public injecting and/or using alone in bathrooms given people space and time to test and inject drugs. Notably, OPSs help mitigate stigma and shame while increasing opportunities for building trust and relationships. These impacts were primarily achieved through shifts in the inner setting and inspite of ongoing challenges in the outer setting including financial, spatial and temporal challenges.

Distrust of health services is inextricably linked to stigma [42], shame [43] and criminalization. Low threshold services are dependent on the achievement of trust and the development of trust is a precondition for accessing services and a mediating factor in service effectiveness [44, 45]. In our study, we heard from participants how the provision of a safe place to use drugs can reduce the stigma and shame of drug use, and facilitate relationships of trust. As in other settings [46], we found that experiential workers played an important role in the social dynamics of the spaces and the enactment of trust. Rance and Fraser [47] and others [48] note the need to ensure the emotional impacts of harm reduction services, including consumption services, do not go unnoticed in evaluations and assessments. With trust recognized as a key to delivering effective care to people who use drugs, trust should be incorporated as a service delivery measure [42]. While trust was facilitated by peers and important for facilitating people to attend OPSs, connections to other health and social services were limited resulting in a missed opportunity to enhance access into other systems of care. This could readily be rectified through inclusion of nurses and social workers as a resource to OPSs.

OPSs were established to witness injections and respond to overdoses. An unintentional impact described by both people accessing, and providing services, was the social function of the space for people who use drugs. OPS, like needle exchanges, function as "safe havens" [49, 50]. This finding reflects what Rance and Fraser [47] have termed "accidental intimacy" which refers to the relationships that can develop in these settings when people inject drugs in the presence of staff. However, relationships, trust, and intimacy is not inherent in the provision of space, and the mechanisms that facilitate such relationships, intimacy and trust require deliberate attention to be realized. For example, other research on supervised injection services identified the limiting effects of services that do not reflect well drug culture and practices of service users [11, 51], as well as the medicalization and hypersanitary approach to service design and delivery [52, 53]. The ability to provide safe and welcoming space was facilitated by the community-led process of developing OPSs [25]. However, as other authors point out, these spaces remain highly gendered and racialized, pointing to the need for culturally and gender appropriate OPSs [54]

The resource of time provided to service users was often described as allowing for safer injecting practices compared to rushed, concealed injections in criminalized and public settings. However, the function of time was also described as providing not only a space to prepare and inject drugs, but also to be drugged. Moore [55] cautions against public health approaches solely focused on reducing the harms, while ignoring the pleasure. Specific to drug consumption services, Duncan [56] asserts the need for consumption rooms to incorporate the consumption experience and pleasure, and not overlook nor consider pleasure to be irrelevant to the public health function. Instead, it is important to acknowledge pleasure in the design and delivery of consumption services and sites.

As long as drug use is prohibited and punished, stigma and shame will continue to be prevalent. Participants described how the provision of time and a space to openly use drugs enabled them talk about their drug use. Harm reduction services, such as needle distribution services, are incomplete, ineffective and even punishing if the service does not provide space to use harm reduction supplies. This is arguably turning people away rather than welcoming them to meet them where they are at [40, 41]. This study and others by the authors, have documented how washrooms in health and social service agencies can become de facto consumption sites. Although such de facto sites provide greater safety than other settings (alone or in public settings), such settings are limited in responding to overdoses and continue to entrench stigma and shame [57]. Scenarios such as needle distribution to reduce transmission of blood borne diseases, or naloxone to respond to overdose, without a more comprehensive harm reduction approach to addressing the overall harms associated with substance use and potentially experienced by the people accessing services. While we observe the importance of moving from safer supplies to safer spaces, the next step has to be a safer source of substances given that the ongoing emergency of overdose deaths is a consequence of an unsafe and toxic drug supply [58, 59].

## Conclusions

The provision of safer spaces for injection drug use within existing community agencies had multiple impacts including zero deaths; earlier intervention to prevent overdoses and reduced trauma; more comprehensive implementation of harm reduction with the introduction of safer spaces for use, mitigation of stigma, and enhancement of the development of trust and relationships. These impacts were a result of shifts in the inner context, and successes were achieved in spite of limitations related to funding, physical space and ongoing criminalization in the outer context. Thus, highlighting the commitment of experiential and non experiential

staff to implement evidence-based services in the wake of ongoing overdose deaths and the need for a safer supply to mitigate an unsafe drug supply.

## Supporting information

**S1 Data. Interview guide–service users.**
(DOCX)

**S2 Data. Interview guide–staff.**
(DOCX)

## Acknowledgments

We acknowledge with respect and humility the traditional territory of the W̱SÁNEĆ(Saanich), Lkwungen (Songhees), Wyomilth (Esquimalt) peoples of the Coast Salish Nation on which the University of Victoria is located. We are grateful for the contributions of staff and services users in each of the agencies where the research was conducted. We specifically acknowledge the contributions of Jordan Cooper at Our Place Society.

## Author Contributions

**Conceptualization:** Bernadette Pauly, Bruce Wallace, Jack Phillips, Mark Wilson, Heather Hobbs, Joann Connolly.

**Formal analysis:** Bernadette Pauly, Bruce Wallace, Flora Pagan.

**Funding acquisition:** Bernadette Pauly.

**Investigation:** Bernadette Pauly, Bruce Wallace.

**Methodology:** Bernadette Pauly, Bruce Wallace.

**Project administration:** Bernadette Pauly, Bruce Wallace.

**Resources:** Bernadette Pauly.

**Supervision:** Bernadette Pauly, Bruce Wallace.

**Validation:** Bernadette Pauly, Bruce Wallace.

**Writing – original draft:** Bernadette Pauly, Bruce Wallace, Flora Pagan.

**Writing – review & editing:** Bernadette Pauly, Bruce Wallace, Flora Pagan, Jack Phillips, Mark Wilson, Heather Hobbs, Joann Connolly.

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
