## [Editor Report · Decision Letter 0]

4 Oct 2019

PONE-D-19-16948

Impact of Overdose Prevention Sites During an Overdose Emergency in Victoria, Canada

PLOS ONE

Dear Dr. Pauly,

Thank you for submitting your manuscript to PLOS ONE. After careful consideration, we feel that it has merit but does not fully meet PLOS ONE’s publication criteria as it currently stands. Therefore, we invite you to submit a revised version of the manuscript that addresses the points raised during the review process

en you are ready to submit your revision, log on to https://www.editorialmanager.com/pone/ and select the 'Submissions Needing Revision' folder to locate your manuscript file.

To enhance the reproducibility of your results, we recommend that if applicable you deposit your laboratory protocols in protocols.io, where a protocol can be assigned its own identifier (DOI) such that it can be cited independently in the future. For instructions see: http://journals.plos.org/plosone/s/submission-guidelines#loc-laboratory-protocols

We look forward to receiving your revised manuscript.

Kind regards,

Dimitra Panagiotoglou

Academic Editor

PLOS ONE

**Journal Requirements**

**Reviewers' comments:**

This paper is a qualitative analysis exploring the perceived impacts of three newly implemented overdose prevention sites (OPSs) in Victoria, BC during the overdose emergency. The authors conducted semi-structured interviews with 27 participants (including experiential staff and clients) of the new OPSs and applied the CFIR framework to interpret the interviews.

Major revisions requested:

1. The structural layout of the paper assumes the audience is very familiar with OPSs, SCSs, and SIFs – and introduces concepts before leaving their explanations until much later in the body of the work. For example, although OPSs and SCSs are first introduced as concepts on page 3, it is not until page 4 (page 5 really) where their similarities and differences are described for the sake of the reader. Similarly, on line 95 (page 5) the authors mention that OPSs prevent fatalities. This is not explained until MUCH later when a client mentions that the quieter/slower pace of the OPS allows him/her to test the drug before consuming more of it (something that would not normally happen in a hurried environment thereby likely increasing the risk of overdose). And on page 13 (line 250-1) mention that the OPSs reduced the need for naloxone etc.

2. There is no explanation for why Victoria was selected as the study site.

3. It would have been beneficial to the reader (especially given the brevity of the interviews) if the interview guide was included as an appendix/supplemental material.

4. Although the authors mention who conducted the interviews, it is unclear who transcribed the codes (e.g. was this contracted to an external services from audio-recordings?), who conducted the coding (number of coders, initials of coders), how coding discrepancies were resolved, if there was any procedure to validate coding, and duration of the interview period (how soon after the implementation of the OPSs the interviews were conducted).

Minor changes recommended:

1. Please revise for run on sentences (e.g. lines 55 – 8), re-introducing acronyms (e.g. line 66), missing refs (e.g. line 71), and typographical errors (e.g. line 99)

2. The formatting of the participants should be revised. Although there are only 27 participants, and for example 15 staff, on line 320 OPS staff 16 is mentioned, and on line 412 staff and service user 17 is listed). Similarly, while there are 12 service users/clients, on line 459 service user 4 is listed, and on line 481 service user 24 is listed. If this is a consequence of some staff being clients of the services as well, this needs to be clarified in the writing and in the nomenclature.

3. Relatedly, language in table 1 needs to be more consistent, and the authors might want to consider endnotes or footnotes to provide additional details on some of the titles they include. Additionally, the number of interviewers that fit each category would help clarify the nomenclature used throughout (e.g. line 360)

---

## [Author Response · Author response to Decision Letter 0]

7 Nov 2019

PONE-D-19-16948

Impact of Overdose Prevention Sites During an Overdose Emergency in Victoria, Canada

PLOS ONE

November 4, 2019

Dear Editor: 

Thank you for the feedback and the opportunity to revise our manuscript. We have outlined our response to each of the points raised during the review process below. Our responses to the editor and reviewer feedback are highlighted in yellow below. 

Journal Requirements

 We have formatted the manuscript in accordance with PLOS ONE’s style requirements including correct use of Level 1 and 2 headings, referencing of Tables as well as use of square brackets with Vancouver style references. 

Reviewers' comments:

This paper is a qualitative analysis exploring the perceived impacts of three newly implemented overdose prevention sites (OPSs) in Victoria, BC during the overdose emergency. The authors conducted semi-structured interviews with 27 participants (including experiential staff and clients) of the new OPSs and applied the CFIR framework to interpret the interviews.

Major revisions requested:

1. The structural layout of the paper assumes the audience is very familiar with OPSs, SCSs, and SIFs – and introduces concepts before leaving their explanations until much later in the body of the work. For example, although OPSs and SCSs are first introduced as concepts on page 3, it is not until page 4 (page 5 really) where their similarities and differences are described for the sake of the reader. 

Response: Thank you for this helpful feedback. We have restructured the front end of the paper. After highlighting the issues related to overdose deaths in the first paragraph, we introduce the concept and evidence related to SCS in the second paragraph. This is followed by a description of the history of OPS’s, prevalence of OPSs as well as the limited research to date of OPSs noting similarities and differences to SCSs. In this way, we are putting upfront definitions and current understandings to date re SCSs and OPS to assist audiences who may not be familiar with these concepts as suggested by the reviewer.

Similarly, on line 95 (page 5) the authors mention that OPSs prevent fatalities. This is not explained until MUCH later when a client mentions that the quieter/slower pace of the OPS allows him/her to test the drug before consuming more of it (something that would not normally happen in a hurried environment thereby likely increasing the risk of overdose). And on page 13 (line 250-1) mention that the OPSs reduced the need for naloxone etc.

Response: There is limited research to date on OPS but the number of fatalities prevented has been reported in the province of BC so we did include this in the background to the study. Also, OPS, like SCS are designed with a primary aim of preventing overdose deaths which we also highlighted early in the introduction to the paper. However, there is Littlr research to date on OPSs and how they work to reduce overdose deaths The two points mentioned by the reviewer above relate to how overdose deaths are reduced that emerged from our research findings. Therefore, we did not move this information into the background section of the paper. We do appreciate that OPSs are novel and nimble responses which are not well known outside of the BC. For this reason, in the opening of the paper we have moved up thr discussion of the concepts, purpose, goals, history and prevalence as well as current research re OPS to address reviewers concerns here which we take to be a similar concern to that outlined above. Our study adds to the understanding of how OPSs work to prevent fatalities and this information is part of our findings as the focus of our study was implementation. 

2. There is no explanation for why Victoria was selected as the study site.

Response: Victoria was selected as the study site for three reasons. First, the researchers had a unique opportunity to undertake this research during the early implementation period because they were already engaged and had been undertaking research in this community as community based researchers for almost a decade. Secondly, Victoria has the third highest rates of overdose deaths in the country. Thirdly, Victoria has faced challenges in establishing and expanding harm reduction services even though BC has favourable harm reduction policy. These reasons have been included in the paper under the section entitled, Materials and Methods. 

3. It would have been beneficial to the reader (especially given the brevity of the interviews) if the interview guide was included as an appendix/supplemental material.

Response: We have added the interview guides as supplemental materials. 

4. Although the authors mention who conducted the interviews, it is unclear who transcribed the codes (e.g. was this contracted to an external services from audio-recordings?), who conducted the coding (number of coders, initials of coders), how coding discrepancies were resolved, if there was any procedure to validate coding, and duration of the interview period (how soon after the implementation of the OPSs the interviews were conducted).

Response: WE have clarified that transcribing was done by an experienced transcriber inhouse. AS well we have clarified we did the coding and provided more detail on our coding procedure as well as validation with our community partners. Further, we have clarified that the interviews were conducted during the first 2-3 months of implementation of the OPS over five weeks from March to April, 2017. 

Minor changes recommended:

1. Please revise for run on sentences (e.g. lines 55 – 8), re-introducing acronyms (e.g. line 66), missing refs (e.g. line 71), and typographical errors (e.g. line 99)

We have had an editor review the paper and address the issues identified above. 

2. The formatting of the participants should be revised. Although there are only 27 participants, and for example 15 staff, on line 320 OPS staff 16 is mentioned, and on line 412 staff and service user 17 is listed). Similarly, while there are 12 service users/clients, on line 459 service user 4 is listed, and on line 481 service user 24 is listed. If this is a consequence of some staff being clients of the services as well, this needs to be clarified in the writing and in the nomenclature.

Response: I think the reviewer might be assuming that we are using consecutive numbering for first staff and then service users. This is not the case we are using numbering related to each site. Service users and staff were interviewed and numbered as data was collected in each site. To avoid this issue, we have revisited the use of participant numbers. So, quotes are identified as being from service users, experiential or non experiential staff. 

3. Relatedly, language in table 1 needs to be more consistent, and the authors might want to consider endnotes or footnotes to provide additional details on some of the titles they include. Additionally, the number of interviewers that fit each category would help clarify the nomenclature used throughout (e.g. line 360)

Response: We have added footnotes for the job titles to assist with understanding and consistency. We have identified the number of staff interview participants as one total for reasons of anonymity and confidentiality.

---

## [Decision Letter · Decision Letter 1]

3 Feb 2020

The Impact of Overdose Prevention Sites During a Public Health Emergency in Victoria, Canada

PONE-D-19-16948R1

Dear Dr. Pauly,

We are pleased to inform you that your manuscript has been judged scientifically suitable for publication and will be formally accepted for publication once it complies with all outstanding technical requirements.

With kind regards,

Dimitra Panagiotoglou

Academic Editor

PLOS ONE

Reviewers' comments:

The first round of revisions made for a much clearer and succinct paper. Please conduct a final read through to clean up minor errors in spelling and writing before publication.

For example:

Punctuation: lines 81, 212 and 519

Spelling: lines 457, 528

Word tense: line 171

Extra/incorrect words: lines 137, 351

Awkward sentence break: lines 116-120

Others suggestions:

Consider adding "and" after comma between "service, staff" (line 151)

The numbering of respondents was helpful in demonstrating diversity of responses. Please re-insert but explain the numbering format in a footnote to avoid confusion.

A second reviewer made the following Minor revision requests:

I have just a few remaining issues to consider:

Missing “Introduction” heading

Introduction:

    Line 43: would suggest just using the rate per 100,000 for this sentence, the absolute number of ODs is hard to interpret without context or comparison and then same comment for the next sentence.

The last sentence in the first paragraph about Victoria seems out of place, it will be important to mention why it’s the chosen site but I think that it’s enough to mention BC in the first paragraph and then Victoria later on

Supervised consumption services: this term seems a bit awkward, is there another way to phrase, like SCS are is a package of services including…..for example drug consumption rooms are not services, a room is a physical space

Line 59: would suggest specifying what blood born diseases

Line 62: would be helpful to understand what requirements the federal government has imposed

Line 80: would define “experiential”

Line 106: save for the discussion

Materials and Methods:

    I found the organization of this section a bit confusing. I would suggest that any “results” would be in the results sections- this would include the table of the different programs

May be helpful to have some subheadings: such as design, setting, participants, inclusion and exclusion criteria, theoretical framework, analysis

Discussion

    The first paragraph should be a summary of the results, not summarizing the benefits of safe consumption services

---

## [Editor Report · Acceptance letter]

11 May 2020

PONE-D-19-16948R1 

Impact of Overdose Prevention Sites During a Public Health Emergency in Victoria, Canada 

Dear Dr. Pauly:

I am pleased to inform you that your manuscript has been deemed suitable for publication in PLOS ONE. Congratulations! Your manuscript is now with our production department. 

With kind regards,

on behalf of

Dr. Dimitra Panagiotoglou 

Academic Editor

PLOS ONE